# Silibinin Protects against H_2_O_2_-Induced Oxidative Damage in SH-SY5Y Cells by Improving Mitochondrial Function

**DOI:** 10.3390/antiox11061101

**Published:** 2022-05-31

**Authors:** Fangfang Tie, Yangyang Fu, Na Hu, Honglun Wang

**Affiliations:** CAS Key Laboratory of Tibetan Medicine Research, Qinghai Provincial Key Laboratory of Tibetan Medicine Research, Northwest Institute of Plateau Biology, Chinese Academy of Sciences, Xining 810008, China; fftie@nwipb.cas.cn (F.T.); fuyangyang@nwipb.cas.cn (Y.F.); huna@nwipb.cas.cn (N.H.)

**Keywords:** silibinin, oxidative stress, reactive oxygen species, mitochondrial function, human neuroblastoma SH-SY5Y

## Abstract

Oxidative stress plays a critical role in the pathogenesis of various neurodegenerative diseases. Increasing evidence suggests the association of mitochondrial abnormalities with oxidative stress-related neural damage. Silibinin, a natural flavonol compound isolated from *Silybum marianum*, exhibits multiple biological activities. The present study investigated the effects of silibinin on H_2_O_2_-induced oxidative stress in human neuroblastoma SH-SY5Y cells. Exposure to H_2_O_2_ (750 µM) reduced the viability of SH-SY5Y cells, which was coupled with increased reactive oxygen species (ROS), abnormal cell morphology, and mitochondrial dysfunction. Remarkably, silibinin (1, 5, and 10 µM) treatment attenuated the H_2_O_2_-induced cell death. Moreover, silibinin reduced ROS production and the levels of malondialdehyde (MDA), increased the levels of superoxide dismutase (SOD) and glutathione (GSH), and increased mitochondrial membrane potential. Moreover, silibinin normalized the expression of nuclear factor 2-related factor 2 (Nrf2)-related and mitochondria-associated proteins. Taken together, our findings demonstrated that silibinin could attenuate H_2_O_2_-induced oxidative stress by regulating Nrf2 signaling and improving mitochondrial function in SH-SY5Y cells. The protective effect against oxidative stress suggests silibinin as a potential candidate for preventing neurodegeneration.

## 1. Introduction

Oxidative stress is two sided, where excessive oxidant challenge causes damage to biomolecules. However, maintenance of a physiological level of oxidant challenge, termed oxidative eustress, is essential for governing life processes through redox signaling [1]. Oxidative damage impairs biomolecules such as lipids, proteins, and DNA and is engaged in inflammation, necrosis, and apoptosis of cells. Counteracting the changes induced by oxidative stress is thus a strategy of neuroprotection [2,3]. Oxidative stress results in the accumulation of reactive oxygen species (ROS), including the free radicals that contain a free electron (e.g., the superoxide anion radical and the hydroxyl radical) and the non-radical (e.g., hypohalous acids, hypochlorite, and chloramines) [4,5]. Oxidative stress plays an important role in neurological injury as the brain carries out intense oxidative metabolism to meet its high energy demands [6]. Accumulating evidence demonstrates the involvement of oxidative stress in the development of Alzheimer’s disease (AD), Parkinson’s disease (PD), and other neurodegenerative disorders [7,8,9,10].

As a critical dynamic organelle in neurons, dynamic disabilities and dysfunction of mitochondria are involved in oxidative stress-related neurodegenerative pathologies. It is well-known that ROS play a critical role in the mitochondria-dependent activation of apoptosis [11,12]. Studies during the past decades have demonstrated that almost all aspects of mitochondrial function can be affected by oxidative stress, including decreased activity of the key enzymes of the respiratory chain and changes in mitochondrial dynamics and energy metabolism [13,14,15]. Therefore, increasing the anti-oxidation capacity and improving mitochondrial function may be promising strategies for the prevention and management of neurodegenerative disorders.

There is a lack of ideal therapies that can overcome oxidative stress without causing serious side effects. Synthesis of new chemicals, drug repurposing, and natural product research are three approaches most widely used for the development of new drug candidates [16,17]. Inspiringly, various natural products have been proven to be effective in anti-aging, memory enhancement, and neuroprotection [18,19,20]. Silibinin is an active component of silymarin extracted from the seeds of *Silybum marianum* (cardui mariae fructus) [21]. Silibinin has been suggested to possess bioactivities such as anti-oxidation, anti-cancer [22], and protection against amyloid-related diseases [23]. A study also showed its protection against cognition and memory deficits in rodent models [24]. In addition, silibinin attenuated astrocyte activation in PD mice by inhibiting the ERK and JNK pathways [25]. These findings suggest silibinin as a promising agent for the treatment of neurodegenerative diseases.

The present study investigated the neuroprotective activity of silibinin in an in vitro model of oxidative stress in H_2_O_2_-exposed human SH-SY5Y cells. Our results indicated that silibinin increases mitochondrial membrane potential and mitochondrial biogenesis and facilitates the scavenging of ROS. These finding suggested silibinin as an antioxidant agent that improves mitochondrial function.

## 2. Materials and Methods

### 2.1. Chemicals and Reagents

Silibinin, 2′,7′-dichlorodihydrofluorescein diacetate (DCFH-DA), JC-1, dimethyl sulfoxide (DMSO), and H_2_O_2_ (30%) were obtained from Sigma-Aldrich (St. Louis, MO, USA). MitoSOX^TM^ Red was from ThermoFisher Scientific (Rockford, IL, USA). Dulbecco’s modified Eagle medium (DMEM) and fetal bovine serum (FBS) were purchased from Gibco (Carlsbad, CA, USA). Kits for measurement of superoxide dismutase (SOD), malondialdehyde (MDA), and glutathione (GSH) were obtained from Nanjing Jiancheng Bioengineering Institute (Nanjing, China). Bicinchonininc acid (BCA) assay kit and protein lysate buffer were obtained from Beyotime (Shanghai, China). Polyclonal antibodies against nuclear factor 2-related factor 2 (Nrf2, #12721), heme oxygenase-1 (HO-1, #26416), extracellular regulated protein kinases (ERK, #4695, p-ERK, #4370), sirtuin 1 (SIRT1, #2493), dynamin-related protein 1 (Drp1, #8570), optic atrophy 1 (OPA1, #67589), and β-actin (#4970) were obtained from Cell Signaling Technology (Danvers, MA, USA).

### 2.2. Cell Culture and Treatment

The SH-SY5Y human neuroblastoma cells (obtained from ATCC, Manassas, VA, USA) were cultured with DMEM supplemented with 10% FBS and 1% penicillin-streptomycin at 37 °C in a humidified atmosphere of 5% CO_2_. To determine the toxicity of reagents, cells were treated with silibinin at concentrations ranging from 0.01 to 100 μM for 12 h. Control cells were treated with medium without silibinin. For oxidative stress induction, cells were exposed to H_2_O_2_ at concentrations ranging from 25 to 1000 μM for 12 h. Control cells were treated with the medium without H_2_O_2_.

### 2.3. Cell Viability Assay

The MTT assay was performed as described previously [26]. SH-SY5Y cells were seeded into 96-well plates at a density of 5 × 10^4^ cells/cm^2^ and grown at 37 °C for 48 h before further treatment. Different concentrations of silibinin (0, 0.01, 0.1, 1, 10, 20, 40, 60, 80, and 100 μM) and H_2_O_2_ (0, 25, 50, 100, 200, 300, 400, 500, 600, 700, 800, 900, and 1000 μM) were added and incubated for 12 h. After optimization, the amounts of silibinin used in the cytoprotection assay were 1, 5, and 10 μM whereas that of H_2_O_2_ was 750 μM, with an incubation time of 12 h. After incubation, 15 μL of 5 mg/mL MTT was added to each well and incubated for another 4 h. The medium was aspirated, and 150 μL DMSO was added into each well to dissolve the formazan crystals. Absorbance was measured at 490 nm with a microplate reader (BioTek, Winooski, VT, USA).

### 2.4. Measurement of GSH, SOD, and MDA Levels

The SH-SY5Y cells were cultured in 6-well plates at a density of 5 × 10^4^ cells/cm^2^ in the presence of 750 μM H_2_O_2_ and incubated with or without silibinin (1, 5, and 10 µM) for 12 h. The treated cells were dissociated by trypsinization and homogenized by ultrasonication for 5 min on ice. The levels of GSH, SOD, and MDA in the homogenates were determined using commercial kits according to the manufacturer’s protocols. The absorbance of the final solution was measured using a microplate reader (BioTek, Winooski, VT, USA).

### 2.5. ROS Production Assay

The intracellular and mitochondrial ROS were detected using the oxidation-sensitive fluorescent probe (DCFH-DA) and MitoSOX^TM^ Red, respectively [27]. Free-state DCFH-DA exhibits green fluorescence (with excitation at 488 nm and emission at 528 nm), the intensity of which positively correlates with the content of ROS. Once in the mitochondria, MitoSOX^TM^ Red reagent is oxidized by superoxide and exhibits red fluorescence (with excitation at 510 nm and emission at 580 nm) [28]. The SH-SY5Y cells were cultured in 12-well slides at a density of 5 × 10^4^ cells/cm^2^ in the presence of 750 μM H_2_O_2_ and incubated with or without silibinin (1, 5, and 10 µM) for 12 h. After incubation with DMEM containing 10 μM DCFH-DA or 10 μM MitoSOX^TM^ Red for 30 min at 37 °C, the stained cells were washed twice with PBS and visualized using confocal laser-scanning microscopy (Leica, Weztlar, Germany).

### 2.6. Mitochondrial Membrane Potential Assay

Changes in mitochondrial membrane potential (ΔΨm) were analyzed using JC-1, a lipophilic cation that selectively enters mitochondria and reversibly changes color from green to red depending on the membrane potential [29]. Red aggregates and green monomers were monitored with excitation at 535 nm/emission at 590 nm and excitation at 485 nm/emission at 535 nm, respectively. The SH-SY5Y cells were cultured in 12-well slide at a density of 5 × 10^4^ cells/cm^2^ in the presence of 750 μM H_2_O_2_ and incubated with or without silibinin (1, 5, and 10 µM) for 12 h. After incubation in DMEM containing 10 μM JC-1 for 30 min at 37 °C, the stained cells were washed twice with PBS and visualized using confocal laser-scanning microscopy (Leica, Weztlar, Germany).

### 2.7. Western Blot Analysis

The SH-SY5Y cells were cultured in 6-well plates at a density of 5 × 10^4^ cells/cm^2^ in the presence of 750 μM H_2_O_2_ and with or without silibinin (10 µM) for 12 h. The treated cells were washed with cold PBS and homogenized in RIPA buffer containing protease inhibitors. The homogenates were centrifuged at 12,000× *g* for 15 min at 4 °C and the supernatant was collected. Samples of 20 µg proteins were separated on 12% or 8% SDS polyacrylamide gel and transferred to PVDF membrane (Millipore: Billerica, MA, USA). The membranes were blocked with skim milk solution at room temperature for 1 h before incubation with primary antibodies overnight at 4 °C. The membrane was washed twice with TBST buffer followed by incubation with secondary antibody at room temperature for 1 h. β-actin was used as internal control. The bands of target proteins were visualized using the ECL assay kit (Millipore: Billerica, MA, USA), and 5200 Multi Luminescent imaging systems (Tanon, Shanghai, China). The light density of protein bands was analyzed with Image J (NIH, Bethesda, MD, USA).

### 2.8. Computational Modeling

Molecular docking was used to analyze the interaction between silibinin and Nrf2 or SIRT1. The analysis was conducted in the Yinfo Cloud Platform (https://cloud.yinfotek.com/) (accessed on 23 November 2021) and with the Autodock Vina program. The crystal structures of Nrf2 (PDB ID: 7K2F) and SIRT1 (PDB ID: 5BTR) were downloaded from RCSB Protein Data Bank (http://www.rcsb.org/) (accessed on 23 November 2021). The structure of silibinin was downloaded from PubChem database (https://pubchem.ncbi.nlm.nih.gov) (accessed on 23 November 2021). The crystal ligand was separated and used to define the binding pocket. AutoDock Vina program was utilized to perform semi-flexible docking with maximum 9 poses output after internal clustering [30].

### 2.9. Statistical Analysis

Data analysis was performed with GraphPad Prism 7.0 (GraphPad Software, Inc., San Diego, CA, USA). Results were presented as mean ± standard deviation (SD) based on three independent experiments. Difference between groups were analyzed with unpaired Student’s *t*-test (two groups) or one-way ANOVA (three or more groups). *p* < 0.05 was considered statistically significant.

## 3. Results

### 3.1. Effects of Silibinin on Cell Viability in SH-SY5Y Cells

The structure of silibinin is presented in Figure 1a. We first examined the effects of silibinin on the cell viability of SH-SY5Y cells using an MTT assay. SH-SY5Y cells were treated with different concentrations of silibinin for 12 h and the results are shown in Figure 1b. The cell viability did not change by treatment of silibinin at concentrations between 0.01 and 10 μM. However, the cell viability decreased at concentrations of silibinin over 10 μM. We next examined the toxicity of H_2_O_2_ at different concentrations (25–1000 μM), which has been widely used as an inducer of oxidative stress. The viability of SH-SY5Y cells decreased after exposure to H_2_O_2_. In comparison with control cells, 69.19% of 700 μM H_2_O_2_-treated cells were viable, while the number in 800 μM H_2_O_2_-treated groups was <50% (Figure 1c). Based on these findings, 1, 5, and 10 μM of silibinin and 750 μM of H_2_O_2_ were used to investigate the effect of silibinin on H_2_O_2_-induced oxidative damage. The observed morphological changes included the disappearance of neuritis and shrinkage of cell bodies (Figure 1d). Remarkably, the H_2_O_2_ exposure-induced neurite retraction and soma shrinkage were attenuated by treatment with silibinin (Figure 1d). In addition, silibinin (1, 5, and 10 μM) restored the cell viability, compared with cells treated with 750 μM H_2_O_2_ alone (Figure 1e).

### 3.2. Effects of Silibinin on Redox State in SH-SY5Y Cells

Oxidative stress has been thought to trigger and sustain the pathogenesis of H_2_O_2_-induced damage. We examined the effects of silibinin on oxidative stress by determining the contents of MDA and its effect on antioxidant status by determining the activities of SOD and GSH. As shown in Figure 2, H_2_O_2_ treatment increased the contents of MDA compared to control cells. Interestingly, treatment with silibinin (1, 5, and 10 μM) decreased the contents of MDA compared to H_2_O_2_-treated cells. In addition, the levels of SOD and GSH of silibinin-treated cells were dose-dependently increased compared to H_2_O_2_-treated cells. The results suggest that silibinin could restore the redox balance in H_2_O_2_-exposed SH-SY5Y cells.

### 3.3. Effect of Silibinin on ROS Production in SH-SY5Y Cells

To provide further evidence for the protective effect of silibinin against oxidative stress, ROS levels in the cells were detected using the fluorescence probes DCFH-DA and MitoSOX. As shown in Figure 3a,b, H_2_O_2_ induced an increase in intracellular ROS content compared to the control cells. The treatment with silibinin attenuated the high ROS level induced by H_2_O_2_. Moreover, compared to the control cells, the expression of MitoSOX signals was increased in SH-SY5Y cells after exposure to H_2_O_2_, suggesting that the mitochondria could be the origin of the observed ROS production. Treatment with silibinin markedly inhibited the H_2_O_2_-induced mitochondrial-derived ROS generation in SH-SY5Y (Figure 3c,d). Thus, silibinin was capable of attenuating intracellular and mitochondrial ROS production, suggesting that silibinin may be a potent scavenger of free radicals.

### 3.4. Effects of Silibinin on Mitochondrial Membrane Potential in SH-SY5Y Cells

To evaluate the mitochondrial function, mitochondrial membrane potential was detected using JC-1 probe. As shown in Figure 3e,f, H_2_O_2_ induced an increase in JC-1 monomers (green fluorescence) compared to the control cells. However, treatment with silibinin decreased the JC-1 monomers and resumed aggregation (red fluorescence).

### 3.5. Effects of Silibinin on Nrf2 Pathway in SH-SY5Y Cells

To explore the molecular mechanism underlying the effects of silibinin against H_2_O_2_, the redox-related signaling pathways were analyzed. As shown in Figure 4a,b, Western blotting results showed that the levels of Nrf2 and HO-1 were reduced by treatment with H_2_O_2_. Remarkably, treatment with silibinin increased Nrf2 and HO-1 compared to H_2_O_2_-treated cells. In addition, we examined the effects of silibinin on H_2_O_2_-induced ERK1/2 phosphorylation, which is involved in both oxidative stress and cell apoptosis. The results showed that H_2_O_2_ treatment reduced phosphorylated ERK1/2, which was attenuated by silibinin (Figure 4c). These results suggested silibinin alleviates oxidative stress via regulating the Nrf2 pathway and ERK1/2 signaling.

### 3.6. Effects of Silibinin on Mitochondrial Fission/Fusion in SH-SY5Y Cells

Mitochondria are dynamic organelles that undergo continuous fission and fusion cycles. Increasing evidence suggests that structural abnormalities in mitochondria are involved in oxidative stress-related pathologies [31,32]. To determine the contribution of silibinin on H_2_O_2_-induced mitochondrial dynamic changes, we examined the effect of silibinin on the expression level of SIRT1, OPA1, and Drp1. As shown in Figure 4d, the expression of SIRT1 was decreased by H_2_O_2_. Interestingly, treatment with silibinin increased the SIRT1 level compared to H_2_O_2_-treated cells. Additionally, silibinin treatment increased OPA1 and Drp1 levels compared to H_2_O_2_-treated cells (Figure 4e,f). These results suggested the involvement of mitochondrial fission–fusion homeostasis in the antioxidant effect of silibinin.

### 3.7. Interaction of Silibinin with Nrf2 or SIRT1

The molecular docking analysis indicated that silibinin could dock well into the binding sites of Nrf2 and SIRT1, as shown in Figure 5. The estimated free energy of silibinin for its binding with Nrf2 and SIRT1 were −9.4 and −7.6 kcal/mol, respectively. For the interaction between silibinin and Nrf2, hydrogen bonds form at residues Val 608, Thr 560, Val 561, Val 512, and Val 465, and hydrophobic interaction occurs at residues Val 561, Ala 366, and Val 418 (Figure 5a,b). For the interaction between silibinin and SIRT1, hydrogen bonds may exist at residues Glu 230, Arg 446, Leu 228, and Ile 153, whereas hydrophobic interactions are present at residues Ile152, Ile 153, and Leu 228 (Figure 5c,d).

## 4. Discussion

Oxidative stress plays a major role in the damage of the central nervous system. In the present study, silibinin was tested for its protective effect in neuroblastoma cells undergoing H_2_O_2_-induced oxidative stress. To overcome the limitations of using primary cultures of mammalian neurons cells, a well-established model of transformed human neuroblastoma SH-SY5Y cells was utilized [33,34]. To find the best conditions to evaluate the rescue effect of silibinin, we first determined the cytotoxicity of silibinin and H_2_O_2_ on SH-SY5Y cells (Figure 1b,c). Due to the concentration-dependent cytotoxicity observed in silibinin-treated cells, the cytoprotective effect of silibinin was tested at concentrations lower than 10 μM. Previous reports have demonstrated the H_2_O_2_-induced oxidative stress at concentrations lower than 1 mM with an incubation period not greater than 24 h [35,36]. In our study, H_2_O_2_ (700–800 μM) was able to reduce cell viability by about 50%, compared to control cells. Silibinin was tested for its capacity of improving cell viability under oxidative stress. The results indicated that silibinin increased the viability of H_2_O_2_-exposed SH-SY5Y cells (Figure 1e).

H_2_O_2_ is one of the most important ROS generated through oxidative stress. It plays a critical role in the normal function of cells by acting on multiple cell signaling pathways [37]. However, H_2_O_2_ causes oxidative damage to nucleic acids, proteins, and membrane lipids [38]. Increasing studies have shown that scavenging ROS, as well as upregulating the levels of SOD and GSH, can protect SH-SY5Y cells from the H_2_O_2_-induced oxidative damage [39]. Herein, the effects of silibinin on ROS production and antioxidant levels were measured. Treatment with silibinin (1, 5, and 10 μM) significantly increased the levels of GSH and SOD (Figure 2a,b). In parallel, the levels of MDA were reduced after incubation with silibinin in H_2_O_2_-induced SH-SY5Y cells (Figure 2c). Consistent with the changes in the enzymes, detection of intracellular ROS with DCFH-DA probe suggested its increase by H_2_O_2_ and the attenuation by silibinin (Figure 3a,b).

The Nrf2-driven free radical detoxification pathways are important endogenous homeostatic mechanisms. In addition, it was reported that the cellular antioxidant system relies on the Nrf2 dissociation from Keap1 and its subsequent translocation to the nucleus, where the activation of antioxidant genes occurs [40]. Our results demonstrated that Nrf2 and HO-1 protein levels were decreased by H_2_O_2_ exposure, which were partially rescued by silibinin treatment (Figure 4a,b). The mitogen-activated protein kinase (MAPK) cascade is an important intracellular signaling pathway involved in oxidative stress responses [41]. In agreement with other related research [42,43], we found that ERK1/2 MAPK was decreased in H_2_O_2_-exposed SH-SY5Y cells. Treatment with silibinin markedly increased the phosphorylation of ERK1/2 in H_2_O_2_-exposed cells.

Mitochondria are the most important source of ROS generated in mammalian cells and play a crucial role in oxidative stress [44]. H_2_O_2_ is a dominant mitochondrial endogenous ROS which has been associated with many chronic neurodegenerative disorders [37]. In addition, cells treated with exogenous H_2_O_2_ exhibit depolarization of mitochondrial membrane potential, which increase its permeability and Cyt-c release [45]. The fluorescent intensity of MitoSOX Red probe was increased after exposure to H_2_O_2_ and attenuated by silibinin, suggesting the suppression of mitochondrial-derived ROS generation by silibinin (Figure 3c,d). Moreover, we also found the increased mitochondrial membrane potential by silibinin (Figure 3e,f). Impaired mitochondrial dynamics is implicated in neurons under oxidative stress [46]. As shown in Figure 4e,f, treatment with H_2_O_2_ resulted in the imbalance of mitochondrial fission and fusion protein. However, silibinin rescued the mitochondrial fragmentation in H_2_O_2_-induced cells by regulating the expression of mitochondrial fission–fusion-associated proteins (OPA1 and Drp1). Taken together, silibinin may be a potential therapeutic agent to treat oxidative pathologies via improving mitochondrial functions.

Despite a lack of direct measurement of the permeability across the blood–brain barrier, several studies have reported the treatment effects of silibinin on neurological disorders after oral administration. For example, silibinin attenuated the loss of dopamine neurons and hippocampal neuron apoptosis in mice models of Parkinson’s disease [47,48]. Moreover, oral administration of silibinin (200 mg/kg) decreased Aβ deposition and the levels of soluble Aβ1–40/1–42 in the hippocampus by downregulating APP and BACE1 in the brains of APP/PS1 mice [49,50]. These effects observed in vivo suggest that silibinin may cross the blood–brain barrier, either directly or in the form of its metabolites. In addition, an in vitro study has been performed in the Caco-2 cell monolayers, a model of the blood–brain barrier [51]. The analyses of different forms of silibinin suggest that future studies should pay efforts to optimizing the formulation of silibinin to improve its entrance into the brain.

## 5. Conclusions

Our study suggested that silibinin alleviates H_2_O_2_-induced oxidative stress and injury in human neuroblastoma SH-SY5Y cells through regulating Nrf2/HO-1 signaling and improving mitochondrial function (Figure 6). Silibinin may be a potent neuroprotective agent that may benefit the prevention and treatment of neurodegenerative disorders.

## Figures and Tables

**Figure 1 antioxidants-11-01101-f001:**
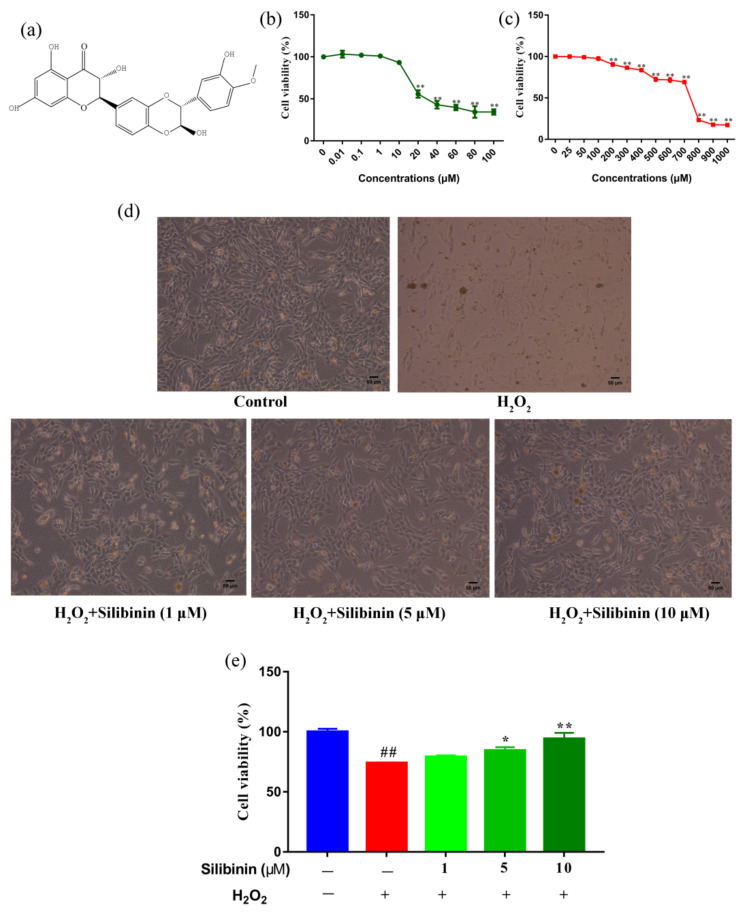
Silibinin protected SH-SY5Y cells from H_2_O_2_-induced cell injury. (**a**) Chemical structure of silibinin. (**b**) The effects of silibinin on the viability of SH-SY5Y cells. Cells were treated with a series of concentration of silibinin for 12 h. (**c**) The effects of H_2_O_2_ on the viability of SH-SY5Y cells. Cells were treated with indicated concentration of H_2_O_2_ for 12 h. (**d**) Morphologic changes of SH-SY5Y cells treated with silibinin (1, 5, and 10 μM) in the presence of 750 μM H_2_O_2_ for 12 h. Scale bar = 50 μm. (**e**) Protective effects of silibinin on SH-SY5Y cells against H_2_O_2_-induced cell injury. Cells were treated with silibinin (1, 5, and 10 μM) in the presence of 750 μM H_2_O_2_ for 12 h. Data are expressed as mean ± SD of three independent experiments (*n* = 3). # *p* < 0.05 and ## *p* < 0.01, compared with the control group; * *p* < 0.05 and ** *p* < 0.01, compared with the H_2_O_2_-treated group.

**Figure 2 antioxidants-11-01101-f002:**
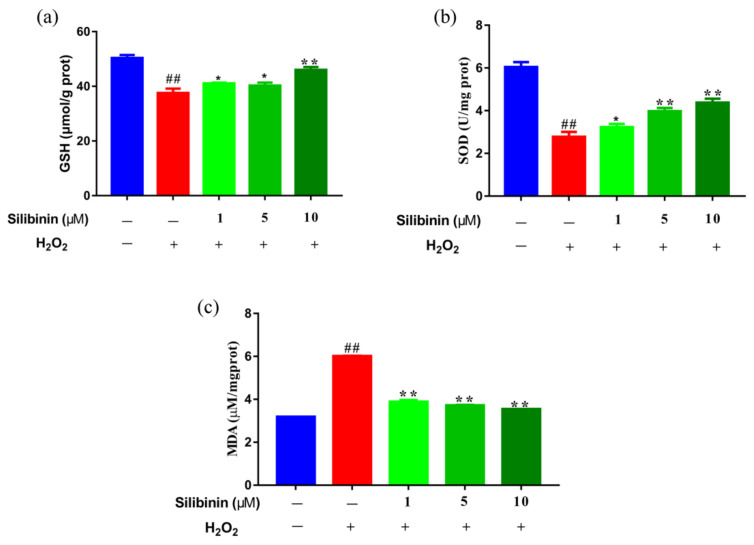
The effects of silibinin on redox state in H_2_O_2_-exposed SH-SY5Y cells. SH-SY5Y cells were treated with different concentrations of silibinin in the presence of 750 μM H_2_O_2_ for 12 h. (**a**) The levels of GSH. (**b**) The levels of SOD. (**c**) The levels of MDA. Data are expressed as mean ± SD of three independent experiments (*n* = 3). # *p* < 0.05 and ## *p* < 0.01, compared with the control group; * *p* < 0.05 and ** *p* < 0.01, compared with the H_2_O_2_-treated group.

**Figure 3 antioxidants-11-01101-f003:**
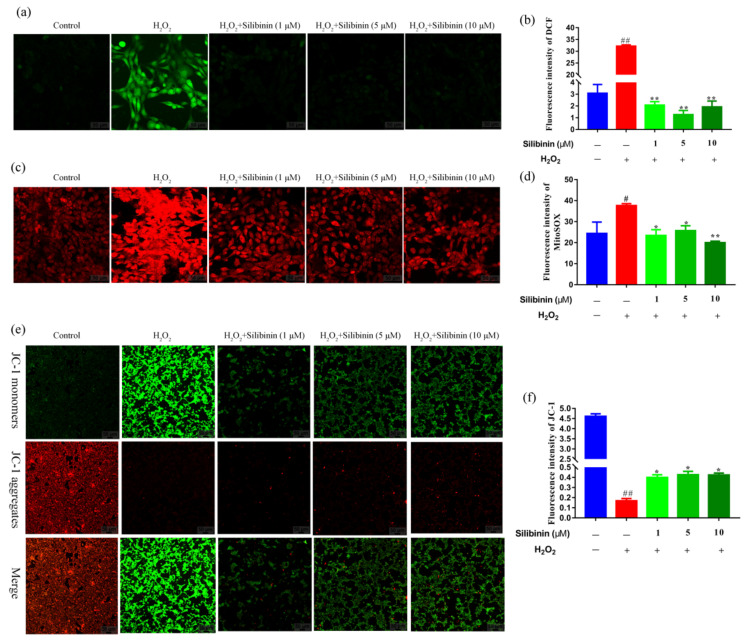
The effects of silibinin on mitochondrial function in H_2_O_2_-exposed SH-SY5Y cells. SH-SY5Y cells were treated with different concentrations of silibinin in the presence of 750 μM H_2_O_2_ for 12 h. (**a**) Confocal images showing fluorescence signal of DCF, scale bar = 50 μm. (**b**) Quantitative analysis of fluorescent intensity in (**a**) by Image J. (**c**) Confocal images showing fluorescence signals of MitoSOX, scale bar = 50 μm. (**d**) Quantitative analysis of fluorescence intensity in (**c**) by Image J. (**e**) Confocal images showing fluorescence signals of JC-1, scale bar = 50 μm. (**f**) Quantitative analysis of fluorescence intensity in (**e**) by Image J. Data are expressed as mean ± SD of three independent experiments (*n* = 3). # *p* < 0.05 and ## *p* < 0.01, compared with the control group; * *p* < 0.05 and ** *p* < 0.01, compared with the H_2_O_2_-treated group.

**Figure 4 antioxidants-11-01101-f004:**
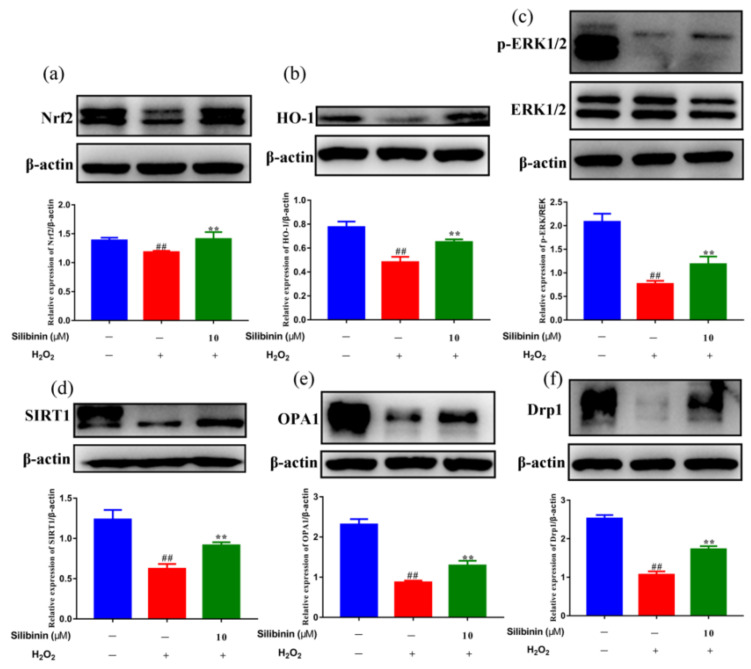
The effects of silibinin on the expression of anti-oxidation-related proteins in H_2_O_2_-exposed SH-SY5Y cells. SH-SY5Y cells were treated with 10 μM silibinin in the presence of 750 μM H_2_O_2_ for 12 h followed by Western blot analysis. (**a**) The expression levels of Nrf2 (100 kDa). (**b**) The expression levels of HO-1 (28 kDa). (**c**) The expression levels of ERK (42, 44 kDa). (**d**) The expression levels of SIRT1 (120 kDa). (**e**) The expression levels of OPA1 (92 kDa). (**f**) The expression levels of Drp1 (83 kDa). Data are expressed as mean ± SD of three independent experiments (*n* = 3). # *p* < 0.05 and ## *p* < 0.01, compared with the control group; * *p* < 0.05 and ** *p* < 0.01, compared with the H_2_O_2_-treated group.

**Figure 5 antioxidants-11-01101-f005:**
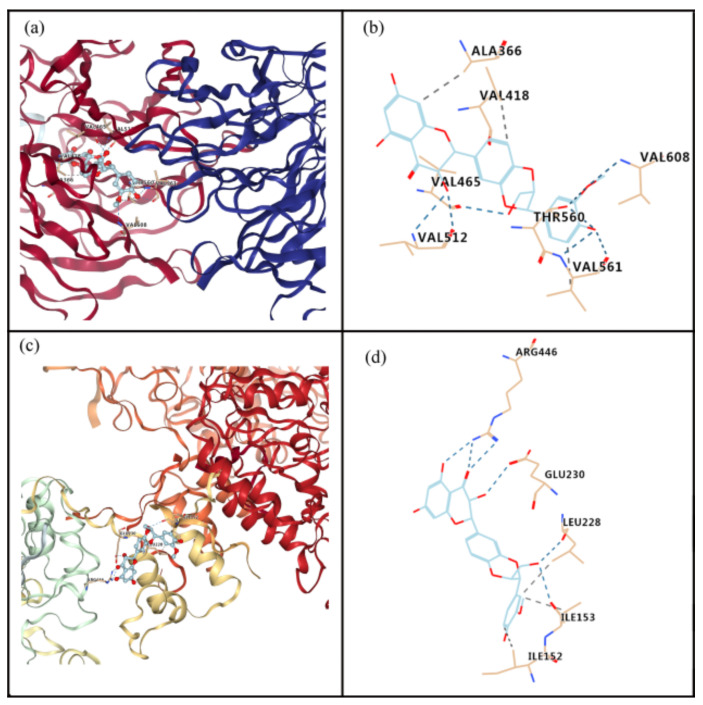
Molecular docking to predict the binding of silibinin to Nrf2 and SIRT1. (**a**) General overview of silibinin docking into Nrf2. (**b**) Demonstration of the predicted binding conformation and corresponding interaction amino acid residues in (**a**). (**c**) General overview of silibinin docking into SIRT1. (**d**) Demonstration of the predicted binding conformation and corresponding interaction amino acid residues in (**c**).

**Figure 6 antioxidants-11-01101-f006:**
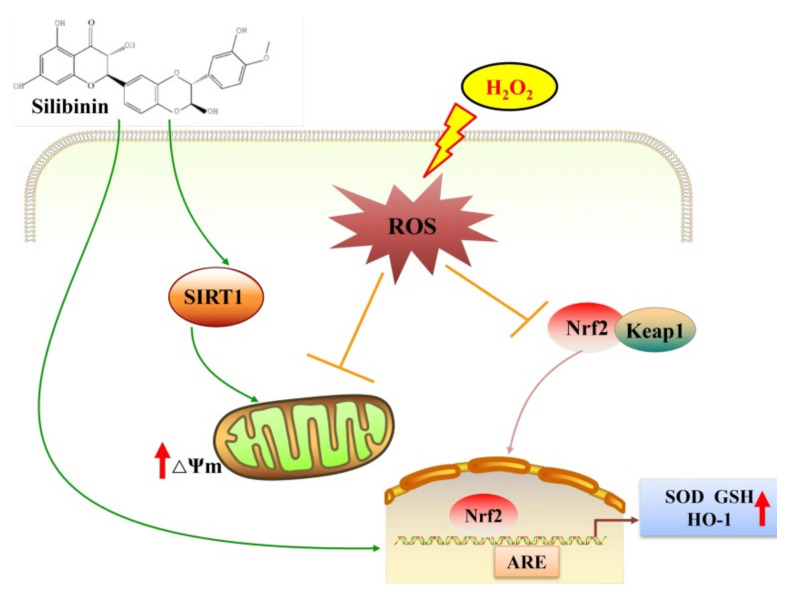
The hypothetical mechanisms for the alleviation of H_2_O_2_-induced oxidative stress in SH-SY5Y cells by silibinin. Excessive H_2_O_2_ leads to intracellular ROS production, with subsequent oxidative stress triggering the Nrf2/HO-1 pathways. Silibinin attenuates H_2_O_2_-induced oxidative stress in SH-SY5Y cells, likely through improving mitochondrial function and upregulating the Nrf2/HO-1 pathways to increase the expression of antioxidant factors.

## Data Availability

All data are available in the manuscript or from Dr. Honglun Wang upon request.

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
