# Peer review of "Silibinin Protects against H2O2-Induced Oxidative Damage in SH-SY5Y Cells by Improving Mitochondrial Function"

_antioxidants, 2022, doi:10.3390/antiox11061101_

Round 1

Reviewer 1 Report

Silibinin was tested for cytoprotective effect in neuroblastoma cells (SH-SY5Y) undergoing H2O2 (700 – 800 μM) induced oxidative stress. They have studied several biomarkers, and conditions. The work is sound and well performed. Their results demonstrate thatNrf2 and HO-1 levels were reduced upon H2O2 incubation and silibinin. They have partially restored the levels. There are some minor mistakes and suggestions to the authors.

Suggestion: Why did they choose the strategy of adding silibinin together with H2O2? Why not use silibinin in a pre-treatment?

  • A more recent concept of oxidative stress should be used. See Sies, H et al. (2017).
  • Correct in some points: enzymes such as SOD and GSH. GSH is not an enzyme, nor MDA.
  • Figure 6. The hypothetical mechanism by which silibinin alleviates H2O2-induced oxidative stress in SH- 322 SY5Y cells is through regulating Nrf2/HO-1 signaling and improving mitochondrial function. The figure does not represent the facts. Explain better.
  • The figure legends should be more detailed.
  • Several references are missing. One of them is shown here: The introduction needs to be updated.https://doi.org/10.1016/j.abb.2018.09.024
  • There are some language mistakes. See annotated pdf, added.

Author Response

Response to the reviewer’s comments

Reviewer 1

Comments and Suggestions for Authors

Silibinin was tested for cytoprotective effect in neuroblastoma cells (SH-SY5Y) undergoing H2O2 (700 – 800 μM) induced oxidative stress. They have studied several biomarkers, and conditions. The work is sound and well performed. Their results demonstrate thatNrf2 and HO-1 levels were reduced upon H2O2 incubation and silibinin. They have partially restored the levels. There are some minor mistakes and suggestions to the authors.

Response: We thank the reviewer for his or her comments that “The work is sound and well performed”. We also appreciate his or her suggestions that improve our manuscript greatly.  

Comments 1: Suggestion: Why did they choose the strategy of adding silibinin together with H2O2? Why not use silibinin in a pre-treatment?

Response: Thanks for the suggestion. According to our previous preliminary experiments, a simultaneous addition of silibinin and H2O2 to the cell culture medium does not cause chemical reactions. This type of treatment was also utilized by a number of studies from other groups [1-4].

Comments 2: A more recent concept of oxidative stress should be used. See Sies, H et al. (2017).

Response: Thanks for the suggestion. We updated the description of oxidative stress to follow the most recent concept in the revised manuscript.

Comments 3: Correct in some points: enzymes such as SOD and GSH. GSH is not an enzyme, nor MDA.

Response: Thanks for the pointing out the problem. We have corrected the mistake by using proper terminology in the revised manuscript.

Comments 4: Figure 6. The hypothetical mechanism by which silibinin alleviates H2O2-induced oxidative stress in SH-SY5Y cells is through regulating Nrf2/HO-1 signaling and improving mitochondrial function. The figure does not represent the facts. Explain better. The figure legends should be more detailed.

Response: Thanks for the constructive suggestion. We have updated figure 6 and revised the legend in revised manuscript.

Comments 5: Several references are missing. One of them is shown here: The introduction needs to be updated. https://doi.org/10.1016/j.abb.2018.09.024

Response: Sorry for the mistake and thank you for pointing them out. The references have been updated in the revised manuscript (Line 58-59).

Comments 6: There are some language mistakes. See annotated pdf, added.

Response: We appreciate the meticulous help. The mistakes have been corrected carefully.

References:

[1] Xiao A, Gan X, Chen R, Ren Y, Yu H, You C. The cyclophilin D/Drp1 axis regulates mitochondrial fission contributing to oxidative stress-induced mitochondrial dysfunctions in SH-SY5Y cells. Biochem Biophys Res Commun. 2017, 483:765-771.

[2] Morán-Santibañez K, Vasquez AH, Varela-Ramirez A, Henderson V, Sweeney J, Odero-Marah V, Fenelon K, Skouta R. Larrea tridentata extract mitigates oxidative stress-induced cytotoxicity in human neuroblastoma SH-SY5Y Cells. Antioxidants (Basel). 2019, 8:427-426.

[3] Zhu A, Wu Z, Meng J, McGeer PL, Zhu Y, Nakanishi H, Wu S. The neuroprotective effects of Ratanasampil on oxidative stress-mediated neuronal damage in human neuronal SH-SY5Y cells. Oxid Med Cell Longev. 2015, 15:1-10.

[4] Du W, An Y, He X, Zhang D, He W. Protection of kaempferol on oxidative stress-induced retinal pigment epithelial cell damage. Oxid Med Cell Longev. 2018, 21:1-14.

Reviewer 2 Report

This study demonstrated that a natural product, silibinin, protects H2O2-induced oxidative stress by modulating mitochondrial function. However, the role of silibinin as an antioxidant (by antioxidase and Nrf2/HO1 pathway) and regulation of mitochondrial function has been well documented, and this study does not add much value.

I have several comments and suggestions:

  • The authors suggest that silibinin binds directly to sirt1 and Nrf2, but there is no data showing that silibinin regulates mitochondrial function by directly binding to sirt1/Nrf2.
  • Line165 figure1b should be figure1c.
  • Figure1d High magnification images are required to support line167's description.
  • In Figure 3, the confocal images are not very impressive, a larger magnification may be better
  • Figure6 Please define MMP. It is unclear whether Nrf2 directly regulates SOD and GSH

Author Response

Response to the reviewer’s comments

Reviewer 2

Comments and Suggestions for Authors

This study demonstrated that a natural product, silibinin, protects H2O2-induced oxidative stress by modulating mitochondrial function. However, the role of silibinin as an antioxidant (by antioxidase and Nrf2/HO1 pathway) and regulation of mitochondrial function has been well documented, and this study does not add much value. I have several comments and suggestions:

Response: We appreciate the reviewer’s suggestions, which improve our manuscript greatly.

Comments 1: The authors suggest that silibinin binds directly to sirt1 and Nrf2, but there is no data showing that silibinin regulates mitochondrial function by directly binding to sirt1/Nrf2.

Response: Thanks for the suggestion. Despite the lack of direct evidence, we have several pieces of data suggest the mechanism to be the binding with SIRT1. First, western blotting results showed that the levels of Nrf2 and SIRT1 were reduced by treatment with H2O2, whereas the treatment with silibinin increased Nrf2 and SIRT1 in H2O2-exposed cells (Figure 4). Second, the docking analysis indicated that silibinin dock well into the binding sites of Nrf2 and SIRT1 (Figure 5). SIRT1 is a member of the Sirtuin family that utilizes NAD+ as a cofactor [1] and multiple studies have suggested the role of SIRT1 in maintaining normal mitochondrial function [2-6]. Combining the information, our study suggests that silibinin alleviates H2O2-induced oxidative stress in human neuroblastoma SH-SY5Y cells, likely through regulating SIRT1signaling to improve mitochondrial function.

We realized that, to determine the requirement of SIRT1, deletions of the SIRT1 gene or blocking of the binding should be performed. We hope the reviewer agrees that these analyses are out of the scope of the present study and could be included in our future studies.

Comments 2: Line165 figure1b should be figure1c.

Response: Sorry for the mistake, it has been corrected in the revised manuscript.

Comments 3: Figure1d High magnification images are required to support line167's description.

Response: Good point. High-resolution images for the cell morphology have been included in the revised manuscript.

Comments 4: In Figure 3, the confocal images are not very impressive, a larger magnification may be better

Response: This is a good point. High-resolution confocal images have been included in the revised manuscript.

Comments 5: Figure6 Please define MMP. It is unclear whether Nrf2 directly regulates SOD and GSH

Response: Thanks for the constructive suggestion. We have updated figure 6, revised the definition of MMP, and added information about the relationship between Nrf2, SOD, and GSH in the revised manuscript.

References:

[1] Imai S, Armstrong CM, Kaeberlein M, Guarente L. Transcriptional silencing and longevity protein Sir2 is an NAD-dependent histone deacetylase. Nature. 2000, 17:795-800.

[2] Pfanner N, Warscheid B, Wiedemann N. Author Correction: Mitochondrial proteins: from biogenesis to functional networks. Nat Rev Mol Cell Biol. 2021, 22:367.

[3] Monsalve M, Wu Z, Adelmant G, Puigserver P, Fan M, Spiegelman BM. Direct coupling of transcription and mRNA processing through the thermogenic coactivator PGC-1. Mol Cell. 2000, 6:307-316.

[4] Cunningham JT, Rodgers JT, Arlow DH, Vazquez F, Mootha VK, Puigserver P. mTOR controls mitochondrial oxidative function through a YY1-PGC-1alpha transcriptional complex. Nature. 2007, 450:736-740.

[5] Li W, Cao J, Wang X, Zhang Y, Sun Q, Jiang Y, Yao J, Li C, Wang Y, Wang W. Ferruginol Restores SIRT1-PGC-1α-Mediated Mitochondrial Biogenesis and Fatty Acid Oxidation for the Treatment of DOX-Induced Cardiotoxicity. Front Pharmacol. 2021, 12:773834.

[6] Scarpulla RC. Metabolic control of mitochondrial biogenesis through the PGC-1 family regulatory network. Biochim Biophys Acta. 2011, 1813:1269-1278.

Reviewer 3 Report

In this paper authors evaluated the putative protection of silibinin in a model of oxidative stress evoked by H2O2 in undifferentiated SH-SY5Y cell line. The toxicity was evaluated through the MTT assay and further western blots were conducted in order to look into the putative intracellular pathways of protection elicited by silibinin. The evaluation of the neuroprotective effects of silibinin in neurons is not a new subject, and there are previous reports on the subject.

The manuscript has several pitfalls that need to be addressed before publication.

Major concerns

-       In figures 2 and 4, the results from the effect of silibinin alone are missing and must be shown to prove that silibinin protects against H2O2 evoked oxidative stress.

-       The results from figure 3, conducted with fluorescent probes must be quantified in graphs and not just representative microphotographs. Again silibinin alone is missing.

Minor Concerns

-       2.3 Cell viability up to 2.7 Western Blot – the number of cells/cm2 must be placed when considering the number of cells that were placed per well. It’s impossible to know how many cells were in the well stating the density/mL, one doesn’t know the volume seeded. 

-       2.7 Western Blot – Some WB images are not perfect, but the major problem are the multiple bands associate to a single protein in some images. Therefore, it must be stated in the methods which bands (Kda) were include for the analysis and quantification. 

-       Figures 1b/c – Statistics is missing in graphs b and c.

-       Figures 1d – The scale bar is missing in the photos.

-       Figure 2 – LDH is not a parameter of oxidative stress and should be removed from this panel of graphs.

Discussion -  The results need to be further discussed and not just mentioning again the results. There are several papers evaluating silibinin protection in SH-SY5Y cells. Also discuss whether the concentrations of silibinin (1-10uM) could be reached in the human brain (or even animal data) following oral intake. 

Author Response

Response to the reviewer’s comments

Reviewer 3

Comments and Suggestions for Authors

In this paper authors evaluated the putative protection of silibinin in a model of oxidative stress evoked by H2O2 in undifferentiated SH-SY5Y cell line. The toxicity was evaluated through the MTT assay and further western blots were conducted in order to look into the putative intracellular pathways of protection elicited by silibinin. The evaluation of the neuroprotective effects of silibinin in neurons is not a new subject, and there are previous reports on the subject. The manuscript has several pitfalls that need to be addressed before publication.

Response: We appreciate the reviewer’s suggestions, which improve our manuscript greatly.

Comments 1: In figures 2 and 4, the results from the effect of silibinin alone are missing and must be shown to prove that silibinin protects against H2O2 evoked oxidative stress.

Response: Thanks for the suggestion. The experiments contained five groups, the control group, model group (750 μM H2O2), and groups of 750 μM H2O2 plus different concentration of silibinin (1, 5 and 10 μM). To show the protective effect of silibinin, we think the strategy should be a comparison between the H2O2-exposed cells without silibinin treatment and that with silibinin treatment. This type of experimental design has actually also been used for the analyses of many other bioactive chemicals [1-6]. In addition, studies have shown that silibinin alone in normal cells does not induce toxic responses [7, 8]. We therefore did not have a silibinin alone group.

Comments 2: The results from figure 3, conducted with fluorescent probes must be quantified in graphs and not just representative microphotographs. Again silibinin alone is missing.

Response: Thanks for the constructive suggestion. Quantitative results have been added into Figure 3 and relevant changes have been made in the text. For the silibinin alone group, please see the addressment for comment 1.

Comments 3: 2.3 Cell viability up to 2.7 Western Blot – the number of cells/cm2 must be placed when considering the number of cells that were placed per well. It’s impossible to know how many cells were in the well stating the density/mL, one doesn’t know the volume seeded.

Response: This is a good point. This information of number of cells/cm2 have been added in the revised manuscript.

Comments 4: 2.7 Western Blot – Some WB images are not perfect, but the major problem are the multiple bands associate to a single protein in some images. Therefore, it must be stated in the methods which bands (Kda) were include for the analysis and quantification.

Response: Thanks for the constructive suggestion. The information about the correct bands have been included in revised Figure legends.

Comments 5: Figures 1b/c – Statistics is missing in graphs b and c. Figures 1d – The scale bar is missing in the photos.

Response: Sorry for the omission and mistake. They have now been corrected in the Figure 1b-c.

Comments 6: Figure 2 – LDH is not a parameter of oxidative stress and should be removed from this panel of graphs.

Response: This is a great point. As suggested, we removed the data of LDH from Figure 2.

Comments 7: Discussion -The results need to be further discussed and not just mentioning again the results. There are several papers evaluating silibinin protection in SH-SY5Y cells. Also discuss whether the concentrations of silibinin (1-10uM) could be reached in the human brain (or even animal data) following oral intake.

Response: Thanks for the constructive suggestion. We have included the suggested points in the “Discussion” of the revised manuscript.

References:

[1] Li M, Fan Y, Zhong T, Yi P, Fan C, Wang A, Liu J, Xu Y. The protective effects of vernicilignan A, a new flavonolignan isolated from Toxicodendron vernicifluum on SH-SY5Y cells against oxidative stress-induced injury. Fitoterapia. 2019, 134:81-87.

[2] Morán-Santibañez K, Vasquez AH, Varela-Ramirez A, Henderson V, Sweeney J, Odero-Marah V, Fenelon K, Skouta R. Larrea tridentata extract mitigates oxidative stress-induced cytotoxicity in human neuroblastoma SH-SY5Y Cells. Antioxidants (Basel). 2019, 8:427-426.

[3] Zhu A, Wu Z, Meng J, McGeer PL, Zhu Y, Nakanishi H, Wu S. The neuroprotective effects of Ratanasampil on oxidative stress-mediated neuronal damage in human neuronal SH-SY5Y cells. Oxid Med Cell Longev. 2015, 15:1-10.

[4] Du W, An Y, He X, Zhang D, He W. Protection of kaempferol on oxidative stress-induced retinal pigment epithelial cell damage. Oxid Med Cell Longev. 2018, 21:1-14.

[5] Zhang H, Yuan B, Huang H, Qu S, Yang S, Zeng Z. Gastrodin induced HO-1 and Nrf2 up-regulation to alleviate H2O2-induced oxidative stress in mouse liver sinusoidal endothelial cells through p38 MAPK phosphorylation. Braz J Med Biol Res. 2018, 51:e7439.

[6] Zhao B, Wang Z, Han J, Wei G, Yi B, Li Z. Rhizoma Paridis total saponins alleviate H2O2‑induced oxidative stress injury by upregulating the Nrf2 pathway. Mol Med Rep. 2020, 21:220-228.

[7] Guo H, Wang Y, Liu D. Silibinin ameliorats H2O2-induced cell apoptosis and oxidative stress response by activating Nrf2 signaling in trophoblast cells. Acta Histochem. 2020, 122:151620.

[8] Gomes VJ, Rezeck Nunes P, Haworth SM, Sandrim VC, Peraçoli JC, Peraçoli MTS, Carlström M. Monocytes from preeclamptic women previously treated with silibinin attenuate oxidative stress in human endothelial cells. Hypertens Pregnancy. 2021, 40: 124-132.

Round 2

Reviewer 2 Report

The authors have addressed my questions and concerns.

Author Response

Response: We thank the reviewer for his or her comments. We are glad that he or she is satisfied with our responses and said that “The authors have addressed my questions and concerns”.

Reviewer 3 Report

In the revised version, the authors addressed most of the concerns, and the manuscript has improved. Sill one major issue remains.

Major concerns

-       In figures 2,3 and 4, the results from the effect of silibinin alone are missing. Authors did not provide this data, which is in my view an important control. It is up to them and the Editors the decision to accept the manuscript with this data.

Minor Concerns

Methods – the paper used 3 different cell densities in the wells for different assays, to compare results the same cell density should be used among different assays.

Discussion -  authors did not address the issue of whether the concentrations of silibinin (1-10uM) could be reached in the human brain (or even animal data) following oral intake. 

Author Response

Comments and Suggestions for Authors

In the revised version, the authors addressed most of the concerns, and the manuscript has improved. Sill one major issue remains.

Major concerns

Comments 1: In figures 2, 3 and 4, the results from the effect of silibinin alone are missing. Authors did not provide this data, which is in my view an important control. It is up to them and the Editors the decision to accept the manuscript with this data.

Response: We appreciate the constructive advice. However, we respectfully have some different thoughts on that “silibinin alone… is an important control”. Firstly, our present study is focusing on the effects of silibinin in pathological conditions (i.e., excessive oxidative stress). In contrast, the “effect of silibinin alone” means its effect under healthy or physiological conditions, which is more like a different topic than that of the pathological conditions. Secondly, to add the group of “silibinin alone” to Figures 2, 3 and 4 is time consuming and will heavily delay the publish of the manuscript. We sincerely hope the reviewer may agree to put this part of study in our work in the future.

Minor Concerns

Comments 2: Methods – the paper used 3 different cell densities in the wells for different assays, to compare results the same cell density should be used among different assays.

Response: Thanks for pointing it out. The cell densities have now been corrected in the revised manuscript.

Comments 3: Discussion - authors did not address the issue of whether the concentrations of silibinin (1-10uM) could be reached in the human brain (or even animal data) following oral intake.

Response: This is a greatly suggestion, we are sorry for a failure of making the expression clear. We added more information on this topic in the “Discussion” and copied here. “Despite a lack of direct measurement of the permeability across the blood-brain-barrier, several studies have reported the treatment effects of silibinin on neurological disorders after oral administration. For example, silibinin attenuated the loss of dopamine neurons and hippocampal neuron apoptosis in mice of Parkinson’s disease [47, 48]. Moreover, oral administration of silibinin (200 mg/kg) decreased Aβ deposition and the levels of soluble Aβ1-40/1-42 in the hippocampus by downregulating APP and BACE1 in the brains of APP/PS1 mice [49, 50]. These effects observed in vivo suggest that silibinin may cross the blood-brain-barrier, either directly or in the form of its metabolites. In addition, in vitro study has been performed in the Caco-2 cell monolayers, a model of the blood-brain-barrier [51]. The analyses of different forms of silibinin suggest future studies should pay efforts to optimizing the formulation of silibinin to improve its entrance into the brain”.

References

[47] Liu, X.; Wang, C.; Liu, W.; Song, S.; Fu, J.; Hayashi, T.; Mizuno, K.; Hattori, S.; Fujisaki, H.; Ikejima, T. Oral Administration of Silibinin Ameliorates Cognitive Deficits of Parkinson's Disease Mouse Model by Restoring Mitochondrial Disorders in Hippocampus. Neurochem Res. 2021, 46, 2317-2332.

[48] Liu, X.; Liu, W.; Wang, C.; Chen, Y.; Liu, P.; Hayashi, T.; Mizuno, K.; Hattori, S.; Fujisaki, H.; Ikejima, T. Silibinin attenuates motor dysfunction in a mouse model of Parkinson's disease by suppression of oxidative stress and neuroinflammation along with promotion of mitophagy. Physiol Behav. 2021, 1, 113510.

[49] Bai, D.; Jin, G.; Zhang, D.; Zhao, L.; Wang, M.; Zhu, Q.; Zhu, L.; Sun, Y.; Liu, X.; Chen, X.; Zhang, L.; Li, W.; Cui, Y. Natural silibinin modulates amyloid precursor protein processing and amyloid-β protein clearance in APP/PS1 mice. J Physiol Sci. 2019, 69, 643-652

[50] Duan, S.; Guan, X.; Lin, R.; Liu, X.; Yan, Y.; Lin, R.; Zhang, T.; Chen, X.; Huang, J.; Sun, X.; Li, Q.; Fang, S.; Xu, J.; Yao, Z.; Gu, H. Silibinin inhibits acetylcholinesterase activity and amyloid β peptide aggregation: a dual-target drug for the treatment of Alzheimer's disease. Neurobiol Aging. 2015, 36, 1792-1807.

[51] Pérez-Sánchez, A.; Cuyàs, E.; Ruiz-Torres, V.; Agulló-Chazarra, L.; Verdura, S.; González-Álvarez, I.; Bermejo, M.; Joven, J.; Micol, V.; Bosch-Barrera, J.; Menendez, J.A. Intestinal Permeability Study of Clinically Relevant Formulations of Silibinin in Caco-2 Cell Monolayers. Int J Mol Sci. 2019, 20, 1606-1617.
